# Research on Sustainable Closed-Loop Supply Chain Synergy in Forest Industry Based on High-Quality Development: A Case Study in Northeast China

**Shuya Wang and Xinjia Tian ***

School of Economics and Management, Northeast Forestry University, Harbin 150036, China
* Correspondence: a1119598081@nefu.edu.cn

**Abstract:** Forestry construction is a fundamental issue related to sustainable economic and social development. However, the traditional development of forestry is currently facing the dilemma of insufficient resource supply, rising labor cost, and a low return on forestry investment, which needs to be transformed into high-quality forestry development. The general view is that synergistic development is an important way to achieve high-quality development. Based on this, the strategic planning and behavioral choices of forestry enterprises under synergistic development are explored. With the objectives of minimizing cost and environmental impacts, and maximizing social benefits, a multi-objective sustainable closed-loop supply chain network planning model is developed to study a real case of a forestry supply chain in northeastern China. A robust optimization approach is used to deal with the relevant uncertain parameters, and a weighted generalized epsilon-constraint approach is applied to solve the multi-objective problem, based on which a GA genetic algorithm is used to solve it. Based on the synergistic development perspective, four different scenario assumptions and sensitivity analyses are made to examine the multi-objective calculation results in this closed-loop supply chain network, and then to analyze the strategic decisions and specific measures for forest industry enterprises to achieve high-quality development. The model results show that realizing supply chain synergy is an effective way to achieve efficient business development in the future. Managers should also establish an acceptable balance between sustainability dimensions while focusing on building a collaborative supply chain development model, where small economic benefits can be appropriately ignored to reduce the harmful effects of the production process on the environment.

**Keywords:** supply chain; chain synergy; multi-objective optimization; high-quality development; forest industry



## 1. Introduction

High-quality development has become the focus and hot spot of research in recent years; compared with other fields, research on high-quality development in forestry is relatively lacking in China [1]. Sustainable development is defined as "meeting the needs of the present without compromising the ability of future generations to meet those needs." To achieve such development, sustainability considers economic, environmental and social aspects simultaneously and seeks to achieve a balance among the three [2]. High-quality development in forestry refers to the basic concept of sustainable development in forestry, with the help of scientific and technological progress and other means, the rational allocation of input factors, the efficient use of resources, and improved production efficiency to achieve sustainable growth [3]. The starting and ending points of high-quality development in forestry are to improve the living environment of forestry-related practitioners, which is also in line with the ultimate goal of China's development [4]. In this context, the high-quality development of the forest industry has become one of the important themes of the times.

Supply chain management is an important component of business operations management, and it has become increasingly popular among academics and related practitioners, becoming a key area of academic and industry interest in recent years [5]. In addition to encompassing the transfer of goods, services, and information from the supplier to the end customer, supply chain management is aiming to continuously improve supply chain performance, and in this systematic process, enhance the sustainability of the customer and the supply chain [6]. Supply chain management has shown a powerful role in improving the management efficiency of enterprises, coping with competitive pressure on resources, and upgrading the industry [7,8]. Therefore, in order to achieve high-quality development in the forest industry, it is necessary to study the design of the supply chain network structure in the forest industry.

High-quality development is the optimization of both quality and quantity, and sustainability is the key to high-quality development [9]. Sustainable development strategies are often linked to supply chain network design and are translated into sustainable supply chain management issues. By solving such problems, the economic, environmental, and social sustainability of a company's production and operation is achieved [10]. The forest industry includes industries such as forest paper, forest board, and furniture products, where the end product can be reused at the end of its life cycle, which is in line with the principles of reverse logistics [11]. The industry's supply chain can therefore also be transformed into a sustainable closed-loop supply chain. High-quality development also needs to consider more aspects on the basis of sustainable development. The value chain is a chain pattern formed by the connection of various activities of an enterprise, whose main objective is to improve the efficiency of production operations and overall competitiveness. The direction of value chain optimization focuses on value chain enhancement activities such as value creation and value addition [12]. The industrial chain is a chain form connected between industries, and its main goal is to improve industrial efficiency and development. The direction of optimizing industrial chains focuses on industrial chain extension activities such as the inter-industry division of labor and collaboration [12]. The synergistic development strategy formed by the interaction of the value chain, industrial chain, and supply chain is an important way and means to accelerate the pace of transformation, upgrading, and quality improvement to achieve high-quality development [13]. Therefore, we can further consider the industry chain and value chain factors based on sustainable supply chains to help enterprises make behaviors and decisions that are in line with the development of the general context, and thus promote the high-quality development of the whole industry.

## 2. Literature Review

The existing literature is described in three aspects: closed-loop supply chain, sustainable supply chain, and sustainable closed-loop supply chain.

### 2.1. Closed-Loop Supply Chain

In recent years, people's awareness of environmental protection has been increasing. At the same time, driven by both environmental laws and regulations, and economic interests, enterprises are paying more attention to being responsible for the whole product life cycle, making a kind of logistics management in the reverse direction of the traditional supply chain, i.e., reverse logistics, an important issue of concern for enterprise managers and researchers. In order to solve the problems of reverse logistics and classical supply chain at the same time, scholars have carried out research work on the closed-loop supply chain (CLSC). Closed-loop supply chains include all elements of forward and reverse logistics, emphasizing the coordination between the two, with the goal of achieving a low-cost, low-emission, and low-consumption product life cycle [14]. CLSC problems have been studied for various industries or industry-specific products such as solar cells [15], tires [16], engine oil [17], cooking oil [18], and communication cables [19]. In the CLSC problem definition, uncertainties and risks in the product logistics process are considered, including uncertain parameters such as the quality and quantity of returned goods, and product

demand [20]. The main research approach to solve the CLSC problem is to design different types of mathematical models and decision methods, such as multi-objective mixed integer linear programming models [21], two-stage mixed integer linear models [19], two-stage multi-period stochastic mixed bilinear optimization models [22], and new methods based on decision trees [23]. The methods for solving the above models mainly include robust optimization [17], fuzzy mathematics-based programming [16,24], etc. Some scholars have discussed the reverse logistics issues in the forest industry, but no scholars have yet conducted a systematic study on closed-loop supply chains in the forest industry. The above CLSC-related research provides an important research basis for determining the network structure in the design of sustainable closed-loop supply chain networks in the forest industry.

*2.2. Sustainable Supply Chain*

With deepening globalization, competitive markets, demand uncertainty, and economic challenges are increasing. The concept of sustainability in supply chains has received more attention [25]. The concept of sustainability shows that the supply chain can only gain a competitive advantage in the market when environmental and social factors are taken into account in the process of improving economic efficiency. Sustainable supply chain (SSC) is the integration of the concept of sustainable development in the whole supply chain, in order to achieve the coordination and optimization of economic, social, and environmental benefits, so as to ultimately achieve the sustainable development of the supply chain [26]. Its related research also involves multiple industries and products. Since the SSC problem needs to consider multiple dimensions, the SSC design model and the method used to solve it are different from CLSC. Based on the triple bottom line optimization model, SSC network design usually includes three dimensions: economic, social, and environmental [27], and some studies have extended it to add more dimensions, such as Baghizadeh's addition of a "minimize the number of lost demands" objective function in the forest supply chain [28]. The forestry supply chain has more research results in this field because of its ecological functions. Campanella et al. conducted a study on forestry in Argentina, and designed a mixed integer linear programming model for determining the location and size of each production facility, and the amount of products and forest residues in order to maximize the total benefits [29]. Edgar et al. designed a multi-objective mixed integer linear programming model for the forest residue biofuel supply chain containing three dimensions [30]. Huang et al. proposed a multi-objective biomass fuel supply chain optimization framework involving multiple stages, using a compromise method to find an economic and environmental balance under raw material and technological constraints [31]. Shaghaygh optimized the supply chain design of forestry biomass based on the risk view of decision makers with the objective of achieving reasonable costs, reducing environmental pollution, and increasing employment opportunities [32]. Arabatzis designed the supply chain of fuelwood based on the demand scenario [33].

The effective implementation of multidimensional SSC problems often relies on multi-objective optimization solving methods. Various model solving methods have also been provided in existing studies, such as genetic algorithms [34], particle swarm algorithms [35], the multi-objective record-to-record travel metaheuristic method [36], weighted objective planning techniques [37], the accelerated Benders decomposition algorithm [38], hybrid robust possibility planning [39], the stochastic fuzzy goal planning method, and the sustainability performance scoring method [40]. The above SSC-related studies provide an important research basis for model setting in the design of sustainable closed-loop supply chain networks for the forest industry.

*2.3. A Sustainable Closed-Loop Supply Chain*

It has been shown that companies in which reverse logistics works closely with the forward supply chain achieve better benefits, which is achieved through sustainable closed-loop supply chain (SCLSC) management [41]. Although SCLSC increases the complexity

of the problem, effectively managed closed-loop logistics not only improves a company's image in front of environmentally concerned customers, but also leads to higher profitability. Meanwhile, the importance of SCLSC is growing, driven by environmental regulations and resource depletion. However, compared with CLSC and SSC, the research in the field of SCLSC is still underdeveloped.

Most of the SCLSC studies that have been presented are generic models that do not take into account specific product or industry characteristics and features. For example, Soleimani et al. studied a SCLSC model that includes multiple recyclables types in reverse logistics, and solved the model and the uncertainty in the model using genetic algorithm and fuzzy algorithm [42]. Rezaei et al. proposed a generic model for SCLSC in various industries and used the cuckoo optimization algorithm to solve the model [43]. Mota, based on the triple bottom line optimization model, designed a SCLSC model that integrates multiple interrelated decisions [10], which was solved by Tautenhain et al. using the Lagrangian relaxation method [44].

Some studies have considered case studies and proposed models for specific industries or products, such as glass [45], LCD TVs [46], tires [47], fluorescent lamps [48], and steel [41], but few studies have been made in relation to the forest industry and its products SCLSC. The forest industry plays an important role in the social and economic development of many regions, and how to increase the economic, environmental, and social benefits of forest industry enterprises has been an important topic in the development of relevant regions. Therefore, it is necessary to make relevant studies on a sustainable closed-loop supply chain of forest industry and its products to meet regional development needs.

### 2.4. Research Gap and Contributions

Most of the previous studies on forestry supply chains have focused on traditional or sustainable supply chains, and closed-loop sustainable supply chains have rarely been considered. For multi-objective models that incorporate environmental dimensions, their environmental objectives are mostly measured using only one factor (mostly carbon emissions). In addition, uncertainty parameters play an important role in supply chain decision making [28]. However, compared to other industries, the uncertainty present in forestry supply chains is rarely mentioned.

To better realize the high-quality development of the forest industry in the new era, and to make up for the above-mentioned research deficiencies, this study takes the paper and paper products manufacturing industry in the forest industry as an example, and studies sustainable closed-loop supply chain management in the forest industry by designing a multi-objective mathematical planning model that includes the economic, environmental, and social aspects of sustainable development. The economic aspect aims at minimizing costs, including labor costs, raw material costs, transportation costs, technology costs, and construction costs; the environmental aspect aims at minimizing environmental impacts, including $CO_2$ emissions and water consumption; and the social aspect aims at maximizing social benefits, mainly measured using the number of jobs created. The results obtained from the model calculations will provide a reference for forest industry enterprises to determine a high-quality development path. The main contributions and innovations of this study are summarized as follows.

(1) A sustainable closed-loop supply chain network and multi-objective mixed integer planning model for the forest products industry was designed and based on a real-life supply chain case study in northeastern China.

(2) Water consumption in the production process is considered in the environmental objectives, rather than carbon emissions alone, as a measure of negative environmental benefits, providing a reference for studies dealing with multiple environmental factors.

(3) The model takes into account the uncertainty of some important parameters in the supply chain network and transforms the uncertainty into a deterministic model using robust optimization methods according to the nature of the parameters.

(4) The SCLSC results were studied under the perspective of synergistic development, according to which suggestions were made for the high-quality development of forest industry enterprises.

Table 1 compares this paper with the existing literature.

**Table 1.** Comparison of this paper with the existing literature.

| Literature | Case Study of SCLSC | Uncertainty | Forestry-Related Industries | Synergistic Development Perspective |
|---|---|---|---|---|
| [9,42–44,46,47] | ✓ | × | × | × |
| [13,15–17,20–24] | × | ✓ | × | × |
| [27] | × | ✓ | ✓ | × |
| [28–32,35,36] | × | × | ✓ | × |
| [40,41,45] | ✓ | ✓ | × | × |
| This paper | ✓ | ✓ | ✓ | ✓ |

## 3. Materials and Methods

The paper and paper products manufacturing industry is an important part of the forest products industry. Paper products are one of the key consumer products used in almost every aspect of our lives. While companies in this industry seek to maximize their own profits, the large amount of water consumed and carbon dioxide emitted during their production process has a significant impact on the environment that cannot be ignored. In addition, the industry's role in improving living conditions, creating jobs, and influencing the pace of regional development has prompted companies and governments to consider not only the economic and environmental aspects of the supply chain, but also the social aspects. It can be noted that paper and paper products manufacturing final products can be reused at the end of their life cycle, in line with closed-loop supply chain principles.

According to the actual situation of China's paper supply chain, papermaking with wood fiber as the raw material has become the development trend of the world paper industry and the realistic requirement for protecting the environment. There are two main lines of transformation from wood resources to paper product circulation: one is the direct use of forest resources as raw materials for paper production, producing a wide range of paper products; the other is to obtain wood fiber waste paper through recovery and recycling, and to process it for pulp and paper production [49]. Based on this, the closed-loop supply chain network structure of paper products designed in this paper is shown in Figure 1.

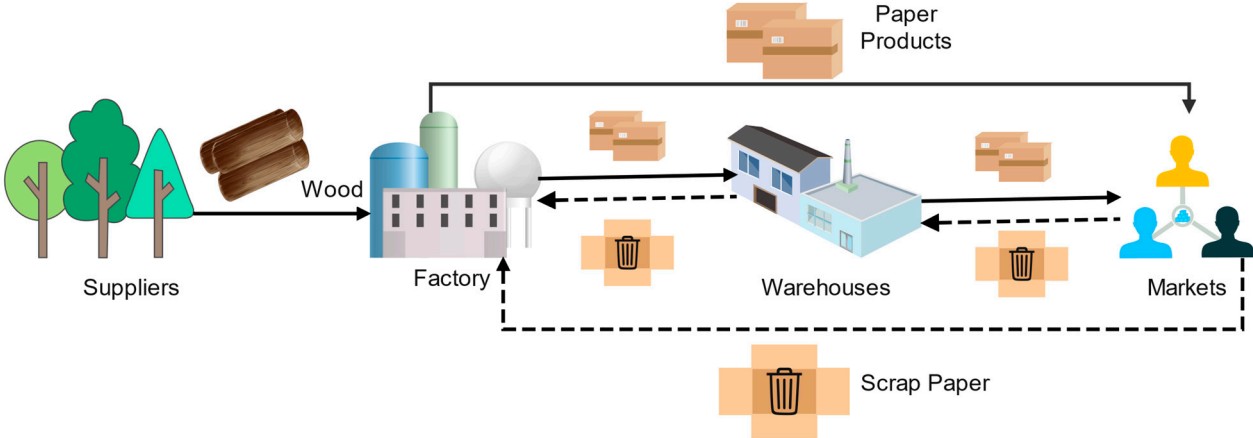

**Figure 1.** Schematic diagram of closed-loop supply chain network structure.

Figure 1 consists of four levels: wood suppliers, paper product processing plants, warehouses and markets, and the transportation process between the four levels. In

the forward logistics, paper raw materials flow from wood suppliers to paper product processing plants to be transformed into final products, i.e., paper products. The final product can enter the warehouse for storage or can go directly to the market for sale. In reverse logistics, end-of-life paper products are recycled in the market and then sent to the factory or directly to the factory via the warehouse. After entering the factory, the scrap paper is processed and will be transformed into the final product again.

*3.1. Mathematical Model*

3.1.1. Parameters

The model needs to determine the parameters and decision variables used before defining the objective function and constraints. The specific symbols and meanings are shown in Tables 2 and 3.

**Table 2.** Parameters of mathematical model.

| Notations | Meaning and Description of the Parameters |
|---|---|
| $sc_{mi}^{max}$ | The maximum supply of product m in supplier i |
| $sc_{mi}^{min}$ | The minimum supply of product m in supplier i |
| $ec_i^{max}$ | Maximum flow rate of entity i |
| $ic_{mi}^{max}$ | Maximum inventory of product m in entity i |
| $ic_{mi}^{min}$ | Minimum inventory of product m in entity i |
| $ea_i^{max}$ | The maximum usable area of the entity i |
| $ea_i^{min}$ | The minimum usable area of entity i |
| $w_i$ | Number of fixed workers required when entity i is selected |
| $lc_i$ | Labor unit price in entity i |
| $wpsq_i$ | Number of workers required per unit area in entity i |
| $wsqmc_i$ | Entity i unit area construction cost |
| $dmd_{mi}$ | The demand for product m in market i |
| $BOM_{mng}^{prod}$ | Bill of materials: Raw material conversion rate in forward logistics |
| $BOM_{mng}^{rem}$ | Bill of materials: Raw material conversion rate in reverse logistics |
| $apu_m$ | Area required for storage of unit product m |
| $apur_m$ | Area required for m storage per unit of recycled product |
| $rmc_{mi}$ | The price of raw material m supplied by supplier i |
| $rpc_m$ | Price of recycled product m |
| $pw_m$ | Weight of product m |
| $sc_m$ | Inventory cost of product m |
| $pc_g^{max}$ | Maximum production capacity of technology g |
| $pc_g^{min}$ | Minimum production capacity of technology g |
| $opc_g$ | Operating costs of technology g |
| $w_g$ | Number of fixed workers required for technology g |
| $tec_g$ | Cost of introduction of technology g |
| $ct_a^{max}$ | Maximum transport capacity of transport mode a |
| $ct_a^{min}$ | Minimum transport capacity for transport mode a |
| $avs$ | Average speed (km/h) |
| $mhw$ | Maximum driving time per week |
| $fct_a$ | Fixed transportation costs for transportation mode a |
| $invt$ | Maximum input for trucks |
| $fp$ | Fuel price (yuan/liter) |
| $w_a$ | Mode of transport, Number of workers required |
| $ei_{mgc}$ | Technology g carbon emissions per unit of product m produced |
| $ei_{ac}$ | Mode of transport, Carbon emissions per kilometer per unit of product m transported |
| $ei_{ic}$ | Entity i carbon emissions per unit area of construction and operation |
| $d_{ij}$ | Distance between entity i and entity j |
| $BigM$ | Upper limit of parameters |
| $yth$ | Number of weeks |
| $wwh$ | Weekly working hours |
| $pc_g^{min}$ | Minimum production capacity of technology g |
| $opc_g$ | Operating costs of technology g |

**Table 3.** Decision variables of mathematical model.

| Notations | Meaning and Description of the Parameters |
|---|---|
| $S_{mi}$ | Inventory quantity of product m in entity i |
| $P_{mgi}$ | Number of products m produced using technology g in entity i in forward logistics |
| $R_{mgi}$ | Number of products m produced using technology g in entity i in reverse logistics |
| $X_{maij}$ | Number of products m transported from entity i to entity j using transport mode a |
| $YC_i$ | Storage capacity of the entity i |
| $YCT_i$ | Entity i needs to use storage |
| $K_{ai}$ | Integer variable, number of transport vehicles for transport mode a in entity i |
| $Q_{aij}$ | Integer variable, the number of round trips to transport products from entity i to entity j using transport mode a |
| $Y_i$ | 0–1 variable, a value of 1 indicates that entity i is selected; otherwise, it is not selected |
| $Z_{gmi}$ | 0–1 variable, a value of 1 indicates that technology g for producing product m in entity i is selected; otherwise, it is not selected |

### 3.1.2. Objective Functions

The model includes three objective functions: economic objective, environmental objective, and social objective.

Economic objective:

$$
\begin{aligned}
min f_{eco} = &\sum_{\substack{(m,i,j)\in FoUTSUPRM \\ (a,m,i,j)\in NetP}} rmc_{mi} X_{maij} + \sum_{\substack{(m,i,j)\in FOUTRP \\ (a,m,i,j)\in NetP}} rpc_m X_{maij} + \sum_{\substack{(a,m,i,j)\in NetP \\ a\in Atruck}} tc_a \cdot 2d_{ij} \cdot Q_{aji} \\
&+ \sum_{\substack{(m,i)\in V \\ (a,m,i,j)\in NetP}} sc_m S_{mi} + \sum_{i\in I_f \cup I_w} w_i \cdot lc_i \cdot wwh \cdot wpt \cdot Y_i + \sum_{i\in I_f \cup I_w} wpsq_i \cdot lc_i \cdot wwh \cdot wpt \cdot YC_i \\
&+ \sum_{\substack{(m,g)\in H \\ i\in I_f}} w_g \cdot lc_i \cdot wwh \cdot wpt \cdot Z_{gmi} + \sum_{\substack{i\in I \\ a\in A_{truck}}} w_a \cdot lc_i \cdot wwh \cdot wpt \cdot K_{ai} \\
&+ \sum_{i\in I_f \cup I_w} sqmc_i \cdot YC_i \cdot (A/P, i, n) + \sum_{\substack{(m,g)\in H \\ i\in I_f}} tec_g \cdot Z_{gmi} \cdot (A/P, i, n) + \sum_{\substack{i\in I \\ a\in A_{truck}}} ftc_a \cdot K_{ai} \\
&\cdot (A/P, i, n)
\end{aligned}
\tag{1}
$$

The economic objective is obtained via cost minimization and consists of 11 items. Items 1–4 are the raw material costs, product recovery costs, transportation costs, and inventory costs, respectively. Items 5–8 are labor costs, Item 5 is fixed labor costs, and the remaining three items correspond to variable costs of labor in entity, technology, and transportation, respectively. Items 9–11 are investment costs, which contain the capital investment in entity, technology, and transportation, where the total investment is converted into annual investment using $(A/P, i, n)$, since the model planning horizon is one year and the time value of money is considered.

Environmental objectives:

$$
min f_{env} = WW \cdot \frac{Z^{water}}{MaxWU} + WG \cdot \frac{Z^{co_2}}{MaxGE}
\tag{2}
$$

Among them,

$$
Z^{water} = \sum_{\substack{(m,g)\in H \\ i\in I_f}} ei_{mgw} \cdot P_{mgi} + \sum_{\substack{(m,g)\in H \\ i\in I_f}} ei_{mgw} \cdot R_{mgi}
\tag{3}
$$

$$
\begin{aligned}
Z^{co_2} = &\sum_{\substack{(m,g)\in H \\ i\in I_f}} ei_{mgc} \cdot pw_m \cdot P_{mgi} + \sum_{\substack{(m,g)\in H \\ i\in I_f}} ei_{mgc} \cdot pw_m \cdot R_{mgi} \\
&+ \sum_{(a,m,i,j)\in NetP} ei_{ac} \cdot pw_m \cdot d_{ij} \cdot Q_{aji} + \sum_{i\in I_f \cup I_w} ei_{ic} \cdot YC_i
\end{aligned}
\tag{4}
$$

The environmental objective is obtained from the minimization of carbon emissions and water consumption in the production process of paper products. The environmen-

tal objective function involves two different dimensions. Therefore, it is normalized in Equation (2) and expressed as a formula with the normalized value multiplied by its given weight. The equation $Z^{water}$ calculates the water consumption for different production technologies. The equation $Z^{co_2}$ calculates the $CO_2$ emissions from technical production, transportation and warehouse construction in the production of paper products. Managers can also adjust the values of WW and WG weights according to corporate development goals.

Social objective:

$$max f_{soc} = \sum_{i \in I_f \cup I_w} w_i \cdot Y_i + \sum_{i \in I_f \cup I_w} wpsq_i \cdot YC_i + \sum_{(m,g) \in H} w_g \cdot Z_{gmi} + \sum_{\substack{(a,i,j) \in Net \\ a \in A_{truck}}} w_a \cdot K_{ai} \quad (5)$$

The social objectives are obtained from the social indicators defined in Equation (5) and are mainly measured by the number of jobs created. The first term reflects the minimum number of workers required for processing plants and warehouses (e.g., administrative staff). Item 2 reflects the number of workers required at different sizes, and the larger the size of the entity, the higher the number required. Item 3 reflects the number of jobs created by each technology. Item 4 indicates the number of jobs created by transportation.

### 3.1.3. Constraints

The model constraints are shown in Table 4.

**Table 4.** Constraints.

| Serial No. | Constraints of Mathematical Model | |
|---|---|---|
| (6) | $\displaystyle\sum_{\substack{j \in I_{sup} \\ a:(a,m,j,i) \in NetP}} X_{maij} = \sum_{(m,g) \in H_{prod}} BOM_{mng}^{prod} P_{ngi}$ | $m \in M_{rm} \wedge i \in I_f$ |
| (7) | $\displaystyle\sum_{\substack{j:(m,j,i) \in F_{INFRP} \\ a:(a,m,j,i) \in NetP}} X_{maij} = \sum_{(m,g) \in H_{rem}} BOM_{mng}^{rem} R_{ngi}$ | $m \in M_{rp} \wedge i \in I_f$ |
| (8) | $\displaystyle\sum_{\substack{j:(m,j,i) \in F_{INCFP} \\ a:(a,m,j,i) \in NetP}} X_{maji} = dmd_{mi}$ | $i \in I_C$ |
| (9) | $\displaystyle\sum_{\substack{a,j:(a,m,i,j) \in NetP \\ (m,i,j) \in F_{OUTSUP}}} X_{maji} \leq sc_{mi}^{max} Y_i$ | $i \in I_{sup} \wedge m \in M_{fp}$ |
| (10) | $\displaystyle\sum_{\substack{a,j:(a,m,i,j) \in NetP \\ (m,i,j) \in F_{OUTSUP}}} X_{maji} \geq sc_{mi}^{min} Y_i$ | $i \in I_{sup} \wedge m \in M_{fp}$ |
| (11) | $\displaystyle\sum_{a,m,j:(a,m,i,j) \in NetP} X_{maji} \leq ec_i^{max} Y_i$ | $i \in I$ |
| (12) | $\displaystyle\sum_{a,m,j:(a,m,i,j) \in NetP} X_{maji} \leq ec_j^{max} Y_j$ | $i \in I$ |
| (13) | $S_{mi} \leq ic_{mi}^{max} Y_i$ | $m \in M_{fp} \wedge i \in \left( I_f \cup I_w \right)$ |
| (14) | $S_{mi} \geq ic_{mi}^{min} Y_i$ | $m \in M_{fp} \wedge i \in \left( I_f \cup I_w \right)$ |
| (15) | $YCT_i = \displaystyle\sum_{maj:(m,a,j) \in NetP} apur_m X_{maji} + \sum_{m:(m,i) \in V} apu_m S_{mi}$ | $i \in I_f \cup I_w$ |
| (16) | $YC_i \geq YCT_{it}$ | $i \in I_f \cup I_w$ |
| (17) | $YC_i \leq ea_i^{max} Y_i$ | $i \in I_f \cup I_w$ |
| (18) | $YC_i \geq ea_i^{min} Y_i$ | $i \in I_f \cup I_w$ |

**Table 4.** *Cont.*

| Serial No. | Constraints of Mathematical Model | |
|---|---|---|
| (19) | $\sum\limits_{a,m,i:(a,m,i,j)\in NetP} X_{maij} \geq Y_j$ | $j \in I$ |
| (20) | $\sum\limits_{a,m,i:(a,m,i,j)\in NetP} X_{maij} \geq Y_i$ | $j \in I$ |
| (21) | $\sum\limits_{m:(a,m,i,j)\in NetP} X_{maij} \leq ct_a^{max}$ | $(a,i,j) \in NetP$ |
| (22) | $\sum\limits_{m:(a,m,i,j)\in NetP} X_{maij} \geq ct_a^{min}$ | $(a,i,j) \in NetP$ |
| (23) | $Q_{aij} \leq BigM{\cdot}Y_i$ | $(a,i,j) \in NetP$ |
| (24) | $Q_{aij} \leq BigM{\cdot}Y_j$ | $(a,i,j) \in NetP$ |
| (25) | $KT_{ai} = \frac{\sum_j 2{\cdot}d_{ij}Q_{aij}}{avs{\cdot}mhw{\cdot}wpt}$ | $(a,i,j) \in Net \wedge a \in A_{truck}$ |
| (26) | $K_{ai} \geq KT_{ai}$ | $a \in A_{truck} \wedge i \in I$ |
| (27) | $\sum\limits_{\substack{a:a\in A_{truck}\\i:i\in I}} ftc_a K_{ai} \leq invt$ | |
| (28) | $K_{ai} \leq BigM{\cdot}Y_i$ | $a \in A_{truck} \wedge i \in I$ |
| (29) | $K_{ai} \leq BigM{\cdot}\sum\limits_{m,j:(a,m,i,j)\in NetP} X_{maij}$ | $a \in A_{truck} \wedge i \in I$ |
| (30) | $P_{mgi} \leq pc_g^{max}{\cdot}Z_{gmi}$ | $i \in I_f \wedge (m,g) \in Hprod$ |
| (31) | $R_{mgi} \leq pc_g^{max}{\cdot}Z_{gmi}$ | $i \in I_f \wedge (m,g) \in Hrem$ |
| (32) | $P_{mgi} \geq pc_g^{min}{\cdot}Z_{gmi}$ | $i \in I_f \wedge (m,g) \in Hprod$ |
| (33) | $R_{mgi} \geq pc_g^{min}{\cdot}Z_{gmi}$ | $i \in I_f \wedge (m,g) \in Hrem$ |
| (34) | $\sum\limits_{g:(m,g)\epsilon Hprod} Z_{gmi} \leq Y_i$ | $m \in M_{fp} \wedge i \in I_f$ |
| (35) | $\sum\limits_{g:(m,g)\epsilon Hrem} Z_{gmi} \leq Y_i$ | $m \in M_{fp} \wedge i \in I_f$ |
| (36) | $P_{mgi}, R_{mgi}, X_{maij}, S_{mi}, YC_i, YCT_i, KT_{ai} \geq 0$ $Y_i, Z_{gmi} \in \{0,1\}$ | |

Equations (6) and (7) indicate that the quantity of raw materials and product flow need to correspond, in forward and reverse logistics. Equation (8) indicates that the products produced by the factory need to meet the market demand. Equations (9) and (10) calculates the supply of raw materials from suppliers in the supply chain network need to meet their own supply limits. Similarly, Equations (11)–(14) indicate that the flow of each entity should satisfy its own flow and capacity limits. Equation (15) calculates the storage that will be used by the plant and each warehouse during the flow of the supply chain network in this paper, which is determined by the current inventory level, and ensuring that it is sufficient to accommodate the incoming flows. Equation (16) indicates that the available area of the entity should meet the storage requirements of the supply chain network. Equations (17) and (18) restrict the physical usable area between the maximum and minimum values of the overall usable area of the warehouse, respectively. Equations (19) and (20) indicate that the entities that assume the roles of consignee and consignor, respectively, will only have flows when they are selected. These two equations can also be considered as minimum flow constraints. If a specific lower bound exists for the parameter, the definition is expressed by multiplying the minimum flow parameter by the variable Yi (as in Equation (14)). Equation (21) ensures that the number of trips between entities multiplied by the capacity of the corresponding transport mode is larger

than the flow between entities. Equation (22) specifies the minimum volume of goods to be transported in the transport mode. Equations (23) and (24) specify the maximum number of transportations, and that the transportations are activated only when the destination is selected. Equation (25) calculates the minimum amount of transportation in the mode of transport, and Equation (26) stipulates that (25) needs to meet the minimum amount. Equation (27) indicates that the truck input should be less than the maximum investment in road transport. Equations (28) and (29) ensure that transport is used only when the entity is selected (when there is a flow). Equations (30)–(35) are technology-related, specifying that the production and remanufacturing technology capacity are between the minimum maximum production level, and also specify that the technology has capacity only when the technology is selected. Finally, Equation (36) specifies the upper and lower bounds for some decision variables.

*3.2. Solution Method*

3.2.1. Robust Optimization

Uncertainty is a part of the business operation that cannot be ignored, so it is very important to consider uncertainty in supply chain networks and improve the reliability of supply chain network design. Uncertainty can be defined as the difference between the information needed to perform a task and the information available [28]. Uncertainty can be divided into two main categories, circumstantial uncertainty and system uncertainty. The first category refers to uncertainties that exist outside of the manufacturing process, such as uncertainties in market demand and supply. The second category is related to uncertainties in the manufacturing process, such as uncertainties in quality, production system failures, product changes, etc. Market demand and material conversion rate are considered as being uncertain in this model. The former belongs to the first category of uncertainty and the latter to the second category. These parameters could be determined in the collection based on historical data or previously accessed data, but due to certain changes, the values of these parameters may increase or decrease, causing the actual values to deviate from the calculated values, affecting the supply chain operation and the completion of the intended targets.

Methods that are commonly used to solve parameter uncertainty include stochastic programming, fuzzy programming, and robust optimization. Stochastic programming is applied to parameters with complete historical data and a specific probability distribution. Fuzzy programming requires the establishment of an affiliation function for uncertain parameters [50], which has advantages in solving uncertainty problems that lack the true values of parameters. Robust optimization requires only that the uncertain set of parameters be closed set and bounded, so that when there are not enough data to estimate the probability distribution of uncertain parameters, it is common to use robust programming to obtain better results for dealing with uncertainty [51]. Therefore, the robust optimization method is used in this paper to consider the uncertainty in the existence of parameters. According to the robust optimization idea [50], the steps for converting the robust counterpart of the model in this paper are as follows.

The general form of the multi-objective planning problem is as follows.

$$
\begin{aligned}
&\min cx \\
\text{s.t.} \quad &Ax \leqslant b \\
&x \in X
\end{aligned}
\tag{37}
$$

The robust corresponding expression for the model with uncertainty parameters in $A$ is

$$\min c'x$$
s.t.
$$\begin{aligned}
&\sum_j a_{ij}x_j + z_i\Gamma_i + \sum_{j\in J_i} p_{ij} \leqslant b_i, \forall i\\
&z_i + p_{ij} \geqslant \widehat{a}_{ij}y_j, \forall i, j \in J_i\\
&-y_j \leqslant x_j \leqslant y_j, \forall j\\
&p_{ij} \geqslant 0, \forall i, j \in J_i\\
&y_j \geqslant 0, \forall j\\
&z_i \geqslant 0, \forall i\\
&x \in X.
\end{aligned} \tag{38}$$

$a_{ij}$ are the elements of matrix $A$; $z_i, y_j, p_{ij}$ are the new variables introduced, $\Gamma_i$ is the degree of uncertainty, $\widehat{a}_{ij}$ is the maximum deviation value, and $J_i$ is the set of uncertain parameters in the ith constraint.

The robust corresponding expression for the presence of uncertainty in $b$ is

$$\begin{aligned}
&\min c'x\\
\text{s.t. } &Ax \leqslant b_i - \Gamma'_i\widehat{b}_i \ \forall i\\
&x \in X
\end{aligned} \tag{39}$$

There are uncertain parameters $BOM_{mng}^{prod}$ and $BOM_{mng}^{renm}$ for the decision variable coefficients in constraints (6) and (7) in this model. $BOM_{mng}^{prod}$ in constraint (6) can be expressed as $\widetilde{BOM_{mng}^{prod}}$, which takes values in the range of $\left[\overline{BOM_{mng}^{prod}} - \widehat{BOM_{mng}^{prod}}, \ \overline{BOM_{mng}^{prod}} + \widehat{BOM_{mng}^{prod}}\right]$, $\overline{BOM_{mng}^{prod}}$, and $\widehat{BOM_{mng}^{prod}}$ denote the scalar (collected data values) and maximum deviation values, respectively. $\widetilde{BOM_{mng}^{prod}}$ is an independent, symmetric, bounded variable that is symmetrically distributed in the range of the above interval centered on the scalar with the mean value. According to the robust optimization Equation (38), constraint (6) robustly corresponds to the constraint as Equation (40).

$$\begin{aligned}
\sum_{\substack{j\in I_{\sup}\\a:(a,m,j,i)\in NetP}} X_{maij} &\geq \sum_{(m,g)\in \text{Hprod}} \overline{BOM_{mng}^{prod}}\cdot P_{ngi} + z_i^1\Gamma_i^1 + \sum_{(m,g)\in \text{Hprod}} P_{ngi}\\
z_i^1 + P_{ngi} &\geq \widehat{BOM_{mng}^{prod}}\cdot y_{ngi}^1\\
-y_{ngi}^1 &\leq P_{ngi} \leq y_{ngi}^1
\end{aligned} \tag{40}$$

Similarly, constraint (7) robustly corresponds to the following constraint.

$$\begin{aligned}
\sum_{\substack{j:(m,j,i)\in F_{INFRP}\\a:(a,m,j,i)\in NetP}} X_{maij} &\geq \sum_{(m,g)\in \text{Hrem}} \overline{BOM_{mng}^{rem}}\cdot R_{ngi} + z_i^2\Gamma_i^2 + \sum_{(m,g)\in \text{Hrem}} R_{ngi}\\
z_i^2 + R_{ngi} &\geq \widehat{BOM_{mng}^{prod}}\cdot y_{ngi}^2\\
-y_{ngi}^2 &\leq P_{ngi} \leq y_{ngi}^2\\
y_{ngi}^1, y_{ngi}^2, &\ z_i^1, z_i^2 \geq 0
\end{aligned} \tag{41}$$

The right-hand side of the equation in constraint (8) contains the uncertain parameter $dmd_{mi}$, which is transformed in the same way as above. The robust counterpart of constraint (8) is given by Equation (42).

$$\overline{dmd_{mi}} - \Gamma_i^3\widehat{dmd_{mi}} \leq \sum_{\substack{j:(m,j,i)\in F_{INCFP}\\a:(a,m,j,i)\in NetP}} X_{maji} \leq \overline{dmd_{mi}} + \Gamma_i^3\widehat{dmd_{mi}} \tag{42}$$

3.2.2. Epsilon-Constraint

Multi-objective optimization problems are usually solved using the idea of simplifying the multi-objective into a single objective. The most commonly used methods to convert multi-objective into single objective are the weight coefficient method and the epsilon constraint method [52]. In this paper, the epsilon constraint method is chosen for the multi-objective treatment. The reasons are as follows.

(1) For linear problems, applying the weighting method to the original feasible domain, the solution results in an angular solution (extreme solution), thus generating only valid extreme solutions. Therefore, when using the weighted method, there may be many weighted combinations to find the same effective extreme solution, which will cost a lot of redundant runs. In contrast, the epsilon constraint method changes the original feasible domain and is able to produce non-extreme valid solutions.

(2) In multi-objective integer and mixed-integer programming problems, weighted methods cannot produce unsupported efficient solutions, while epsilon constraint methods do not suffer from this drawback.

(3) In the weighting method, the scale of the objective function has a great influence on the obtained results. Therefore, before forming the weighted sum, the objective function needs to be quantized. In the epsilon constraint method, this step can be omitted. According to the idea of the epsilon constraint method in the related literature [53], the specific steps of this paper to deal with the multi-objective problem are as follows.

The general expression of the epsilon constraint method is shown in Equation (43).

$$
\begin{aligned}
min\theta_1 f_1(x) - \delta \times \left( \theta_2 \frac{sl_2}{\gamma_2} + \theta_3 \frac{sl_3}{\gamma_3} + \cdots + \theta_p \frac{sl_p}{\gamma_p} \right) \\
\text{s.t. } f_p(x) + sl_p = \varepsilon_p, \forall p \in (2, P)
\end{aligned}
\tag{43}
$$

where $\delta$ is generally taken in the interval $[10^{-6}, 10^{-3}]$, $\theta_p$ is the target weight, $\gamma_p$ is the range of values of the pth objective function, $sl_p$ is the slack (or residual variable) of the corresponding target, and $\varepsilon_p$ is taken in the range of values of the pth objective function. For the multi-objective treatment of the model in this paper, the objective function is transformed into the following form.

$$
\begin{aligned}
min\theta_1 f_{eco}(x) - \delta \times \left( \theta_2 \frac{sl_2}{\gamma_2} + \theta_3 \frac{sl_3}{\gamma_3} \right) \\
\text{s.t. } f_{env}(x) + sl_2 = \varepsilon_2 \\
f_{soc}(x) + sl_3 = \varepsilon_3
\end{aligned}
\tag{44}
$$

3.2.3. Model Solving Algorithms

The solution of the mixed integer programming model is an Np-hard problem, in which small-scale problems can be solved by solvers such as LINGO, GAMS, and CPLEX, but as the problem scales up, the solvers are often difficult to solve, and meta-heuristic algorithms are required. Genetic algorithms are metaheuristics that simulate genetic recombination and evolution, and are widely used for their efficient global search capability, especially to solve complex problems such as network design, path planning, facilities, and site selection [34]. In addition, genetic algorithm, as one of the classical algorithms, is more mature in related research, simple and easy to operate in optimization software, and has a proven effectiveness [54,55]. Therefore, in this paper, the GA algorithm is used to solve the model.

*3.3. Case Study*

A forestry enterprise in northeast China was used as an example to conduct a study on a sustainable closed-loop supply chain network. The following factors were considered in selecting the enterprise.

(1) The scale has reference. The selected enterprise is a small and medium-sized enterprise (SME), which is in line with the scale of the majority of paper enterprises in

China. Small- and medium-sized enterprises account for more than 90% of the total number of enterprises, are the driving force of the national economy and social development, and are vital to stabilizing economic growth, enhancing economic activity, ensuring the integrity of the production system, and stabilizing employment. The study of SMEs is more informative to other enterprises in the industry.

(2) The strategy implementation is feasible. The selected enterprise has favorable conditions to realize the synergistic development of the supply chain, and the enterprise strategy selection problem is consistent with the research problem. The enterprise is located in Heilongjiang province, a major forestry province in China, and the conditions are relatively mature in terms of solving the wood supply problem and achieving industry chain extension such as forest–paper–pulp integration. In addition, the enterprise has the ability to develop and improve production technology, which is an important prerequisite for realizing value-adding in the value chain.

(3) The supply chain network structure is consistent with the studied problem. The business scope of the selected enterprises includes paper products production and sales, waste paper recycling, etc., covering both forward and reverse logistics.

Through a survey of companies, the finalized supply chain network contains three suppliers, two warehouses, three production technologies, and four consumer markets, and the transportation mode is road transport. Combined with the proposed model, the results obtained can be used as a basis for strategic choices and operational decisions by company managers, helping to analyze and discuss the total costs, carbon emissions, supplier selection, warehousing options, product allocation in processing plants, and production technology selection in achieving sustainable development, as well as providing a reference for other companies in the industry.

The location of each node of the case study is illustrated in Figure 2.

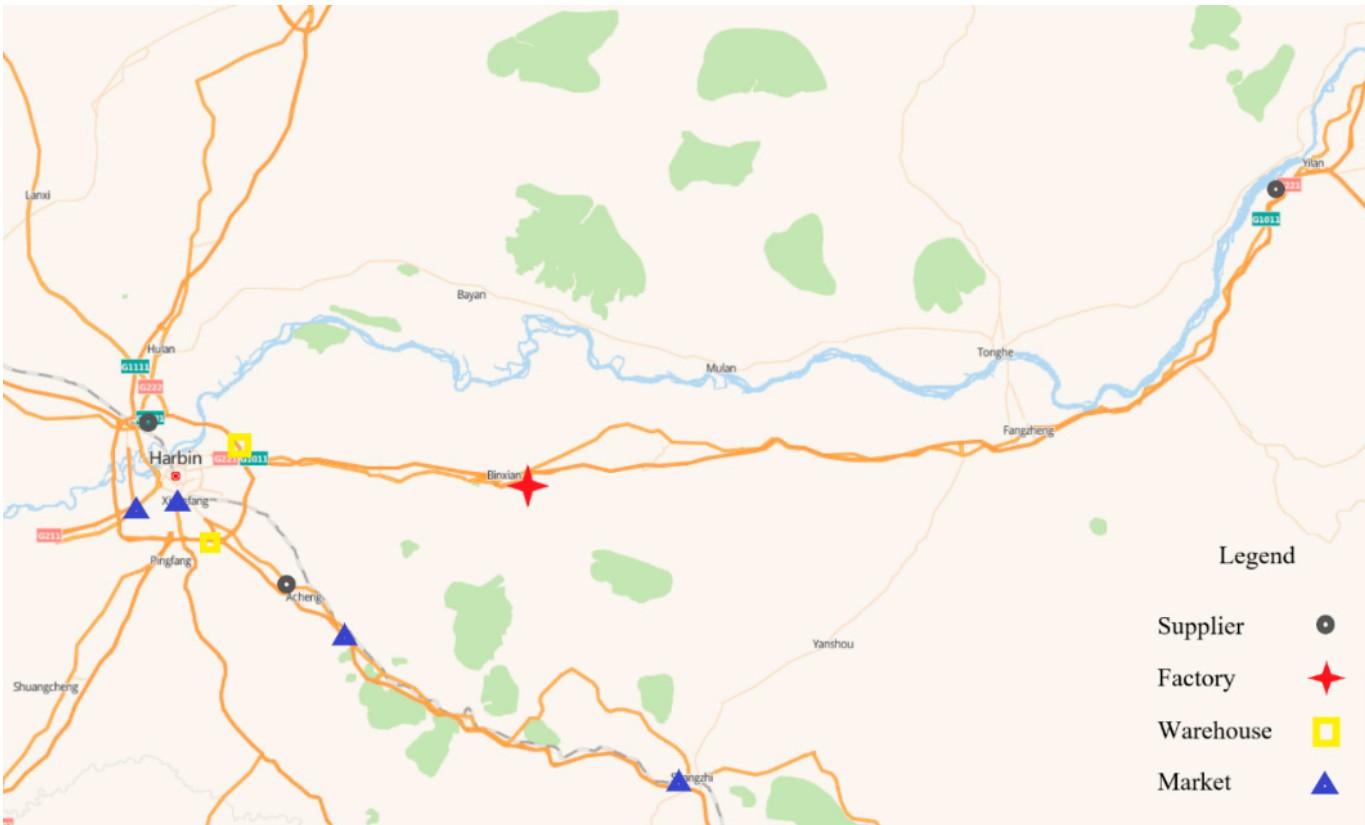

**Figure 2.** The location of different facilities of the case study.

Data were obtained through interviews with corporate managers of the study sample, corporate websites, and corporate annual reports. The main parameters are shown in Tables 5 and 6. The payoff table are shown in Table 7, which is required for the solution of the Epsilon-Constraint method.

**Table 5.** Main parameters of suppliers.

| Parameters | Supplier 1 (m³/year) | Supplier 2 (m³/year) | Supplier 3 (m³/year) |
|---|---|---|---|
| $sc_{mi}^{max}$ | 600,000 | 500,000 | 800,000 |
| $sc_{mi}^{min}$ | 100,000 | 100,000 | 100,000 |
| $ec_i^{max}$ | 1,000,000 | 1,000,000 | 1,000,000 |

**Table 6.** Main parameters of factory and warehouses.

| Parameters | Factory | Warehouse 1 | Warehouse 2 |
|---|---|---|---|
| $ec_i^{max}$ | 200,000 tons/year | 200,000 tons/year | 200,000 tons/year |
| $ic_{mi}^{max}$ | 2800 tons/year | 7692 tons/year | 6000 tons/year |
| $ic_{mi}^{min}$ | 920 tons/year | 2564 tons/year | 2000 tons/year |
| $ea_i^{max}$ | 12,000 m²/year | 10,000 m²/year | 10,000 m²/year |
| $ea_i^{min}$ | 3000 m²/year | 3300 m²/year | 2600 m²/year |
| $lc_i$ | 4000 RMB/month | 6000 RMB/month | 6000 RMB/month |
| $wpsq_i$ | 0.007 person/m²/year | 0.009 person/m²/year | 0.009 person/m²/year |
| $sqmc_i$ | 9431.6 RMB/m²/year | 685.9 RMB/m²/year | 598.42 RMB/m²/year |

**Table 7.** Payoff table.

| | $f_{eco}$ | $f_{env}$ | $f_{soc}$ |
|---|---|---|---|
| $f = f_{eco}^{min}$ | $5.4185 \times 10^8$ ($f_{eco}^{min}$) | 1.9166 | 183.06 |
| $f = f_{env}^{min}$ | $5.4189 \times 10^8$ | 1.5817 ($f_{eco}^{min}$) | 180.70 ($f_{eco}^{min}$) |
| $f = f_{soc}^{min}$ | $5.5721 \times 10^8$ ($f_{eco}^{max}$) | 1.9999 ($f_{eco}^{max}$) | 364.53 ($f_{eco}^{max}$) |

## 4. Results

### 4.1. Solution Results of the Model

According to the payoff table and the epsilon constraint method, $\delta$ is taken as 10-3, $\theta_1$ is taken as 0.4, and $\theta_2$ and $\theta_3$ are taken as 0.3. Due to the presence of the slack (residual) variables, the target weights ($\theta_1$, $\theta_2$, and $\theta_3$) are taken to have insignificant effects on the objective value [53]. In the actual production operation, the decision maker can also choose the appropriate weights more flexibly according to the actual situation. According to Equation (43), $\varepsilon_2$ and $\varepsilon_3$ are calculated and taken as 1.78 and 270, respectively. in this model, two dimensions of the environmental objective function, carbon emissions and water consumption, are considered as equally important, so WW and WG are both taken as 0.5. The coefficients are expanded by the same multiple without affecting the values of the decision variables. For computational simplicity, let WW and WG be taken as 1, respectively.

In summary, the model is solved by invoking the GA function, with Equation (44) as the objective and Equations (40)–(42) and (9)–(36) as the constraints. The solution results for the main variables of the model are shown in Table 8. The solution results represent the specific decisions that a company needs to make to achieve a sustainable supply chain.

**Table 8.** Main results solved by the model.

| Variables | Results | Dimension |
|---|---|---|
| Stock quantity of factory s | 920 | t |
| Stock quantity of warehouse 1 | 2564 | t |
| Stock quantity of warehouse 2 | 2000 | t |
| Warehouse 1 | Choose | - |
| Warehouse 2 | Choose | - |
| Technology 1 | Choose | - |
| Technology 2 | Choose | - |
| Technology 3 | Choose | - |
| Number of Trucks | 36 | Vehicle |
| Volume of products produced using technology 1 | 40,000 | t |
| Volume of products produced using technology 2 | 20,000 | t |
| Volume of products produced using technology 3 | 21,000 | t |
| Market 1 | Choose | t |
| Market 2 | Choose | t |
| Market 3 | Choose | t |
| Market 4 | Choose | t |
| Amount of recycled paper in the factory | 9300 | t |
| Amount of recycled paper in warehouse 1 | 7860 | t |
| Amount of recycled paper in warehouse 2 | 8100 | t |

### 4.2. Results Based on Synergistic Scenario Design

In order to explore the specific synergistic development strategy decisions of enterprises to achieve high-quality development, the model is applied to make four different scenarios of assumptions to examine the results of multi-objective calculations in this closed-loop supply chain network. Scenario 1 (S1) is a traditional sustainable supply chain network (shown in Figure 1). Scenario 2 (S2) and Scenario 3 (S3) are based on Scenario 1, assuming the extension of industry chain and assuming the enhancement of the value chain, respectively. Scenario 4 (S4) assumes the realization of "supply chain synergy". Each scenario parameter is generated based on the changes of existing supply chain network parameters.

A common form of industry chain extension for paper companies is forest–paper–pulp integration. The most direct effect that this model can have is to reduce the price of wood, which in turn affects the cost of raw materials and inventory costs in the supply chain network. Value chain is an economic activity that creates value for the enterprise in the production and operation process. The enterprise value chain enhancement is a dynamic process of adding value to each production link, and the technology-related parameters will change in the context of achieving value chain enhancement. Therefore, based on the traditional sustainable supply chain network, S2 is generated by changing the cost parameters, S3 is generated by changing the technical parameters, and S4 is generated by changing both the cost and technical parameters.

The results of solving the supply chain network model for different scenarios show that the largest share of the economic objective function value is the cost of raw materials, and the largest share of the social objective function value is the number of jobs created by the construction entity. The components of the two objectives remain the same under the change of scenario, so they are not listed here. The extracted information related to the environmental objectives is shown in Figures 3 and 4.

From Figure 3, we can observe the share of carbon emissions generated by the production, construction, and transportation processes of the supply chain, and the specific changes of each share in four different scenarios. According to the figure, in the whole supply chain network, the percentage of carbon emissions from transportation is the highest, product production is the second highest, and construction entity is the least. Among them, the carbon emission share of product production is the highest in S4, the carbon emission share of construction entity is the highest in S1, and the carbon emission share of product transportation is the highest in S2. This indicates that the optimal production

decisions of firms change in different scenarios, and the enterprise's inventory management, product transportation, and product production solutions should be adjusted according to the scenarios.

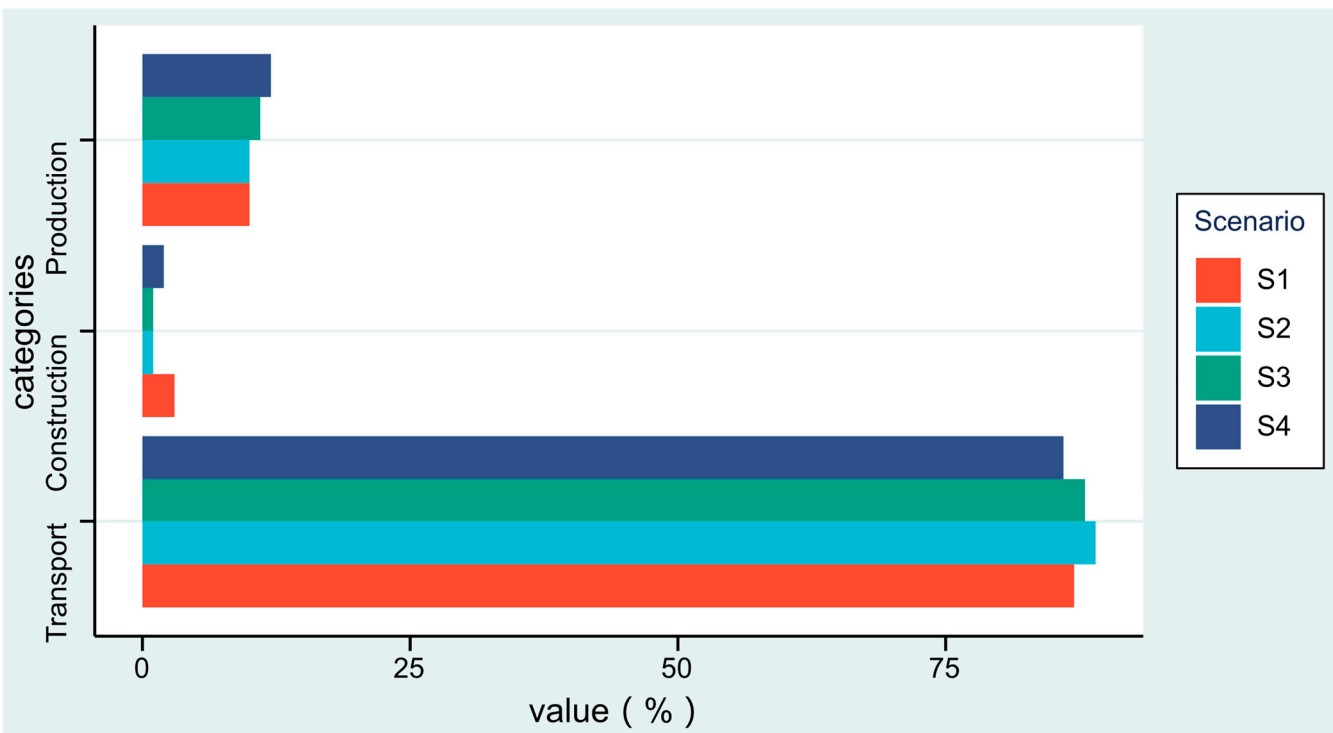

**Figure 3.** Carbon emission of each part.

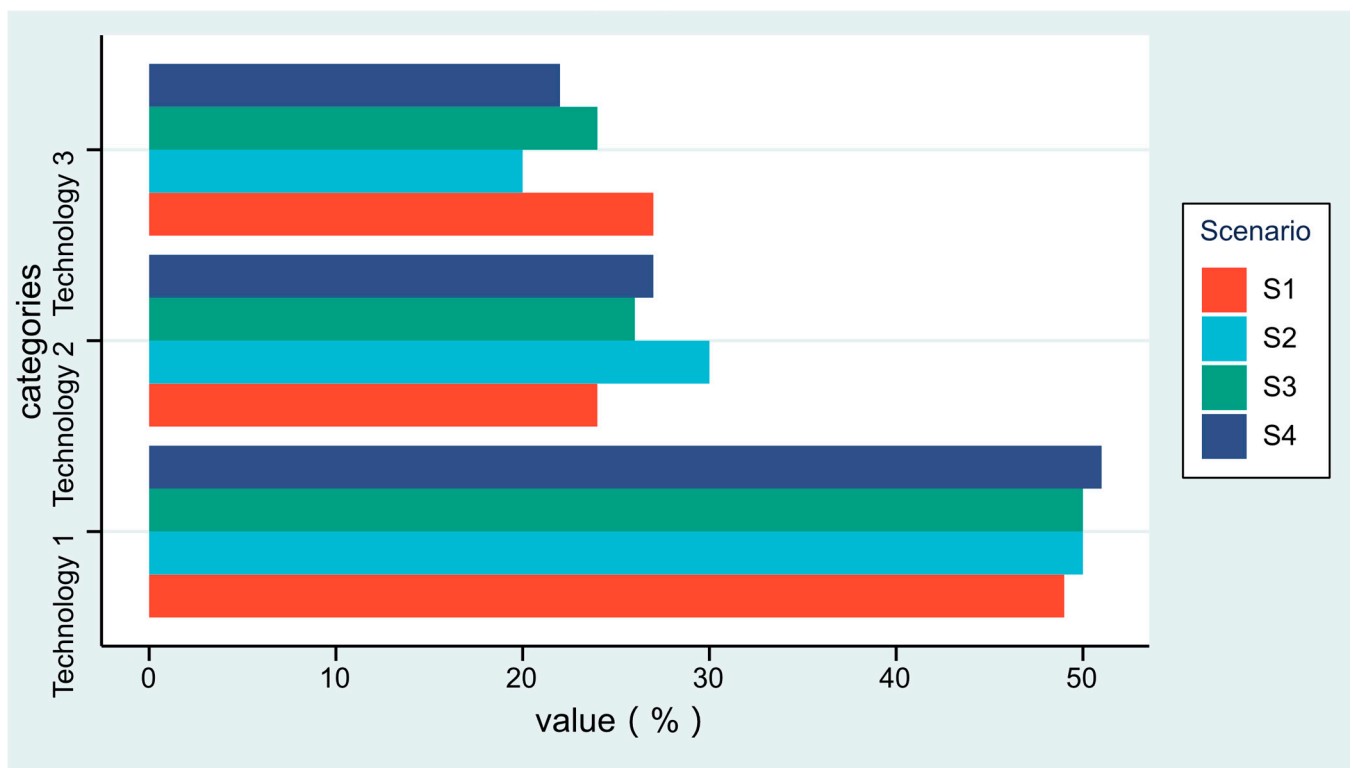

**Figure 4.** Water consumption of each part.

Figure 4 shows the share of water consumption for products produced by different technologies, and the specific variation of each share in four different scenarios. There is a direct relationship between water consumption and product output, so that the frequency of use of each technology line can be observed in Figure 4. As shown in the figure, the most water consumed in the whole supply chain network is the technology 1 production line, and the proportion of technology 2 and technology 3 use changes with the scenario. Compared with S1, the percentage of using technique 2 increased, and the percentage of using technique 3 decreased in S2–S4. In order to consider different potential scenarios and to investigate the influence of these different scenarios on the proposed SCLSC, sensitivity analysis is performed on some key parameters in Section 4.3 and 4.4.

### 4.3. Results of Synergy-Based Dimensional Changes

The general view is that the strategic objectives set by the firm produce better performance in terms of the economic objectives without considering the environmental and social situations. This suggests that the sustainability dimension of the business development strategy has a greater role in influencing the model, and therefore, an uncertainty analysis is performed on the model dimensions. In each of the four scenarios, three cases are considered: first, only the economic aspect is considered (unidimensional), denoted by "Eco"; second, both the economic and environmental aspects are considered (bidimensional), which is the same as the dimension considered in the green supply chain, denoted by "Green"; third, the problem is assumed to be sustainable, which means that environmental, social, and economic aspects are considered (multidimensional), denoted by "Sus". Based on the above settings, the model was run to obtain the calculation results. For the convenience of observation, the values of each objective function are normalized, and the function value of "Sus" under S1 is set to 1. The specific values of the run results are shown in Figures 5 and 6.

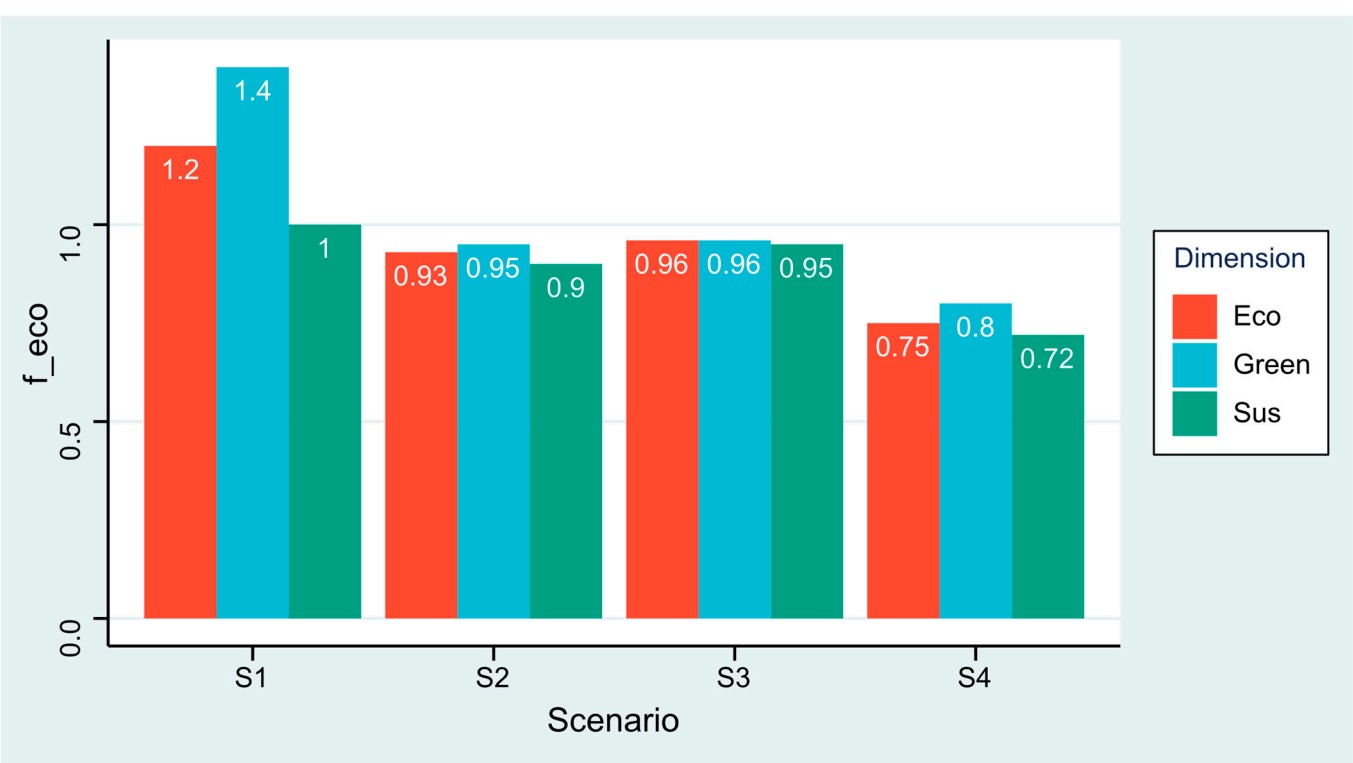

**Figure 5.** Results of the sensitivity analysis of the sustainability dimension of $f_{eco}$.

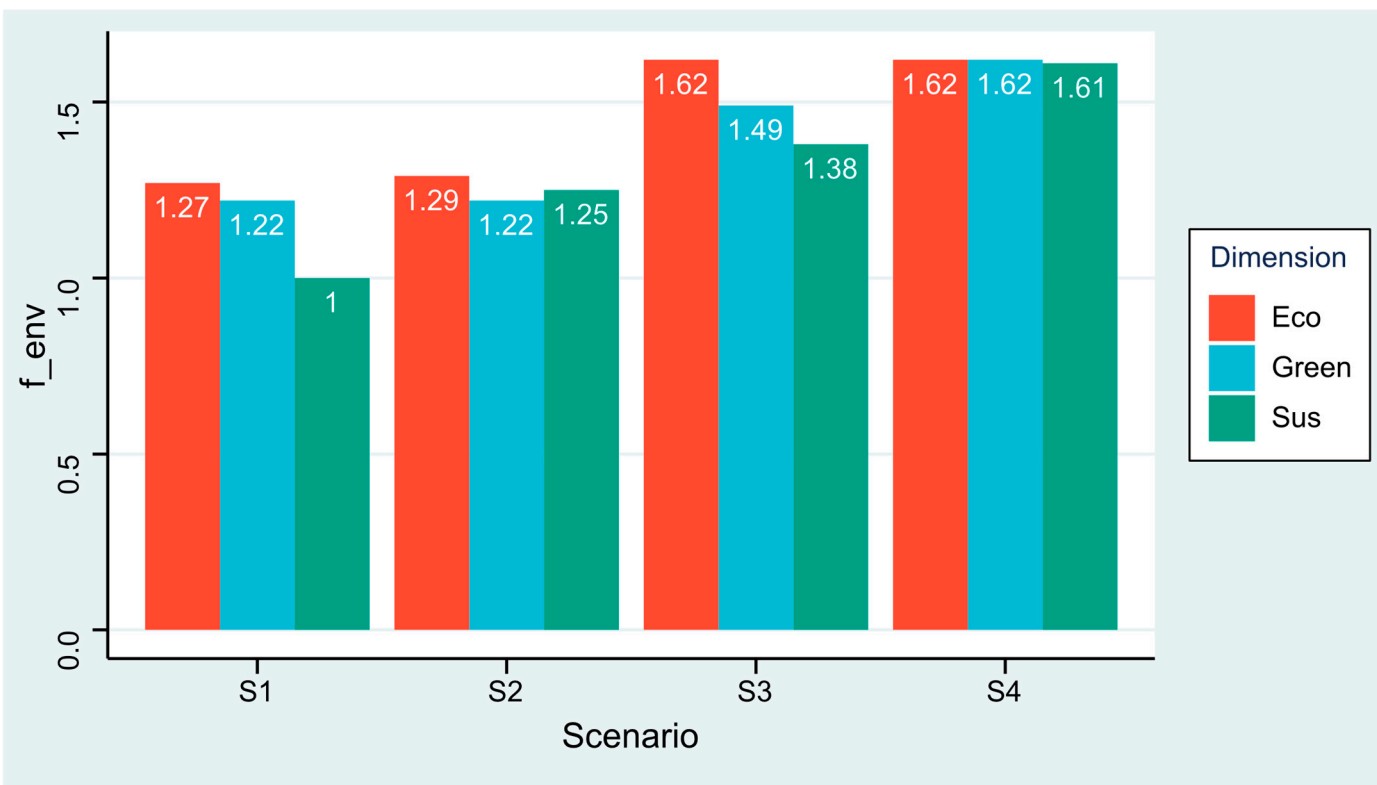

**Figure 6.** Results of sensitivity analysis of sustainability dimensions of $f_{env}$.

Figures 5 and 6, respectively, show the results of the sensitivity analysis of the sustainability dimensions of the economic and environmental objectives for each scenario. It should be added that the value of the social objective function is constant when the dimension is changed, so the results of the analysis for this objective are not presented here. It also shows that social goals are not affected by changes in the sustainability dimension of corporate development strategies.

Combining the results shown in Figures 5 and 6, it can be seen that the values of the economic objective function are lower in all the same situations than in the economic case alone, i.e., considering more dimensions can instead reduce the economic cost. This result does not indicate a contradiction to the general view, but rather a change in the optimal decision solution for the firm under the sustainability goal. Sustainable development requires the coordination of economic, environmental, and social goals, and increasing production increases the economic costs and negative environmental impacts. Taking S1 as an example, the solution results according to Figures 5 and 6 show that in the sustainable (Sus) case, both the economic and environmental objective function values are reduced, and the optimal decision solution for the company is to achieve the best combination of the three dimensions by reducing part of the production. It can also be seen from Figure 6 that the values of the environmental objective function calculated from the unidimensional aspect are higher than those of both the bidimensional and multidimensional aspects in the same scenario, which indicates that by considering more dimensions of strategic objectives, companies are able to make decision options that are beneficial to the environment.

In addition, synthesizing Figures 5 and 6, observing the mean value of the calculation results of each dimension under different situations, and comparing S1, the overall calculation results from S2 and S3 show that when the supply chain network realizes the industrial chain extension or when the enterprise realizes the value chain enhancement, the economic objective function value decreases and the environmental objective function value increases, and the optimal decision solutions of enterprises under S2 and S3 tend to have lower economic costs, because sacrificing small environmental benefits can save larger

costs. From the calculation results of S4, it can be seen that when the supply chain realizes "supply chain synergy", the economic goal is further reduced and the environmental impact is increased, while the calculation results of this scenario are more stable in the dimensional sensitivity analysis. This indicates that "supply chain synergy" can help companies to achieve sustainable development under any strategic objectives.

### 4.4. Results Based on Changes in Price Factors

In supply chain networks, raw material costs and demand are the two categories of variables most characterized by fluctuations. Uncertain demand factors have been taken into account in the model, so a sensitivity analysis is performed on the price factor of recovered waste paper. The fluctuation ranges of the values of each objective function of the model are discussed for $\pm 20\%$ and $\pm 30\%$ fluctuations of waste paper prices under four scenarios, respectively.

As can be seen from Figure 7, the degree of fluctuation in total cost is similar in the four scenarios when the price of waste paper fluctuates positively and negatively. Relative to the other three scenarios, the degree of cost volatility caused by fluctuations in waste paper prices is minimal under S3. As can be seen from Figure 8, the environmental objective function value decreases with the price increase, and the environmental objective function value in S4 is the most stable, but the overall degree of fluctuation is small ($\pm 0.05\%$ or so). As can be seen from Figure 9, the value of the social objective function is largely unaffected by changes in the waste paper price factor, which at the same time reflects that the choice of warehouse and production line options does not change under changes in waste paper prices. From Section 4.3, it can be found that the social objective is measured by the number of jobs, which calculates the number of jobs created in physical operations, production lines and transportation. The number of workers required for transportation is determined by the number of means of transportation, so the social target values fluctuate only when changes in storage centers or technical parameters cause changes in related decisions.

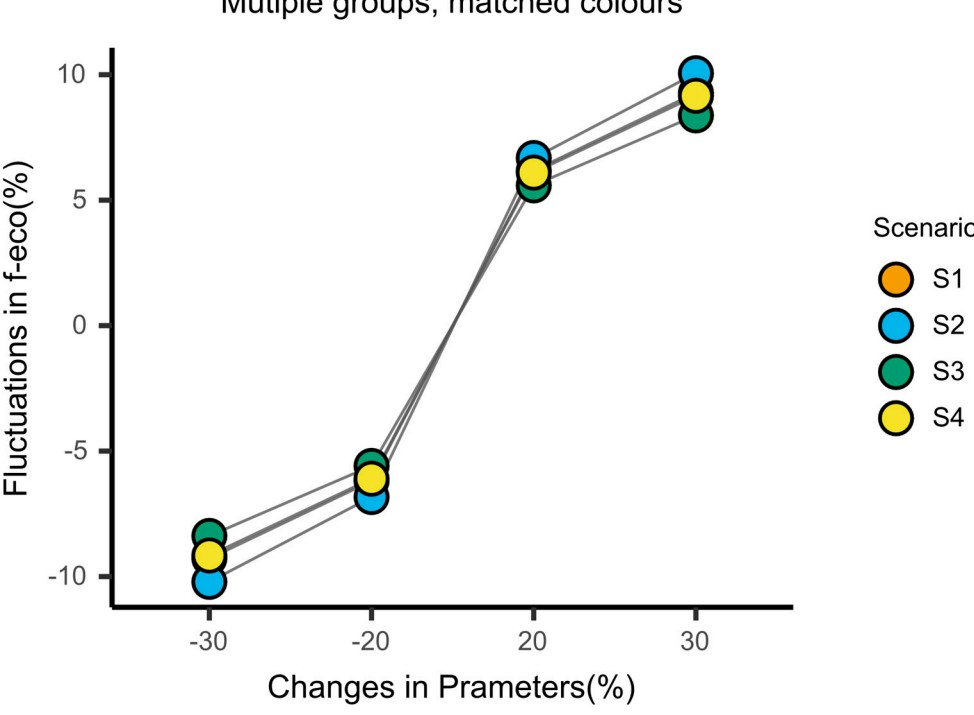

**Figure 7.** Sensitivity analysis of the $f_{eco}$ by changing parameters.

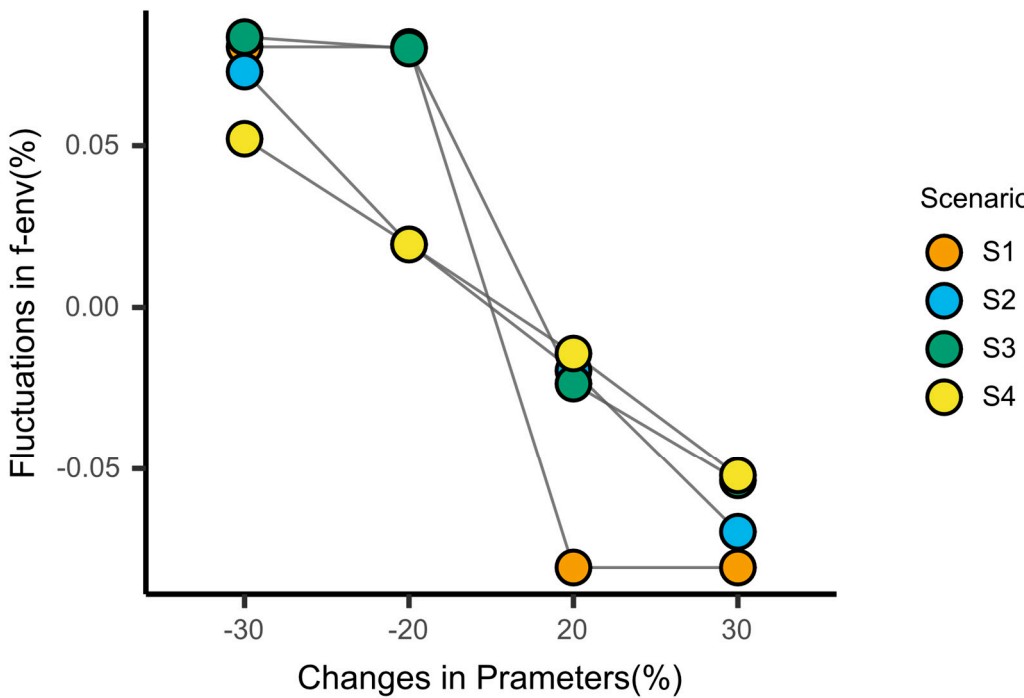

**Figure 8.** Sensitivity analysis of the $f_{env}$ by changing parameters.

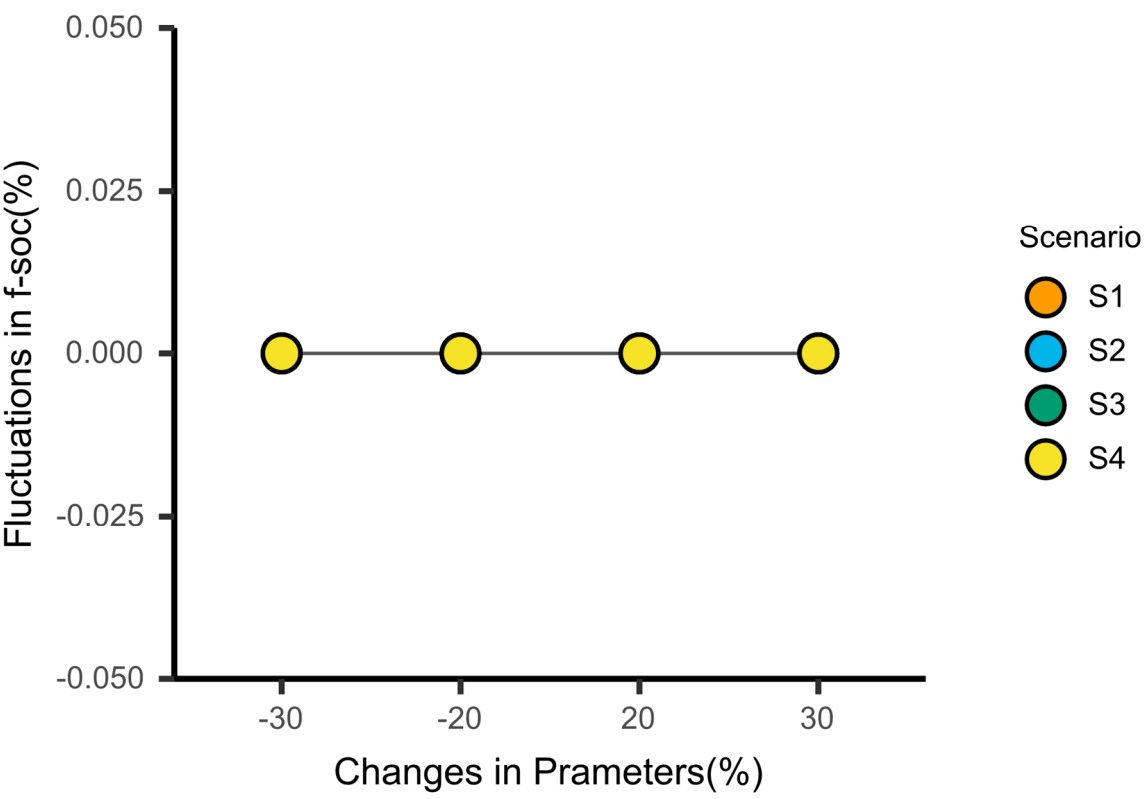

**Figure 9.** Sensitivity analysis of the $f_{soc}$ by changing parameters.

## 5. Discussion

In this paper, the paper and paper products manufacturing industry in the forest industry is taken as the research object, and a multi-objective closed-loop supply chain network model for paper products is established under the conditions of uncertainty in demand and technology, with the objective of minimizing the economic cost, minimizing

the environmental impact, and maximizing social benefits; and the influence of scenario transformation under the different roles of three chains (supply chain, industry chain, and value chain) on the model is considered simultaneously. The results of the model runs provide managers with the following references.

(1) The main cost of the supply chain is the cost of raw materials. Due to the significant impact of this cost, even a small reduction in cost can have a large impact on the final economic efficiency of the entire chain. Owing to the fragmented nature of forest resources, forestry enterprises tend to be more remote in their geographical distribution and are separated from forestry upstream suppliers and consumers, resulting in higher raw material transaction costs. The use of information technology will make communication between forestry enterprises and upstream enterprises more convenient, and big data-based business is conducive to the accumulation of transaction credit, the enhancement of market transaction efficiency, and the reduction in raw material transactions cost, while also saving the cost of contract enforcement and supervision. Business managers can also reduce costs at the source of raw material supply by initiating active cooperation with the government and upstream companies to achieve forest, paper, and pulp integration. Conversely, they can seek the best suppliers and enter into customized contracts with designated suppliers, thereby obtaining discounts in raw material acquisition.

(2) The largest share of carbon emissions is generated by the transportation process. According to the study by the International Energy Agency on the transportation model, the influencing factors of transportation energy consumption can be divided into four categories: activity level, transportation structure, equipment efficiency level, and fuel structure. Combined with the carbon emission measurement method based on activity level equivalence, the above influencing factors correspond to: total transportation, transportation structure, the energy consumption of transportation vehicles, and the carbon emission coefficient of fuels, respectively. According to the characteristics of forestry enterprises, and combined with the emission reduction program in the "Carbon Peak Action Program by 2030" issued by the Chinese State Council, enterprise managers can optimize the transportation structure and energy consumption of transportation vehicles. Managers can increase transportation modes, such as train transportation, air transportation, ship transportation, etc., to achieve transportation structure optimization by finding the balance between environmental impact and transportation cost, and guiding the reasonable distribution of transportation volume among different transportation modes. In addition, business managers can reduce environmental impacts and improve sustainability performance by maintaining transportation tools, reducing transportation losses, and adjusting transportation modes to maintain low transportation tool energy consumption.

(3) The model results show that value chain upgrading can effectively respond to external price changes and reduce risks in the implementation of the firm's decision options. Corporate R&D personnel play an important role in value chain enhancement, especially highly skilled personnel with scientific research capabilities who are key elements for improving the quality of human resources and shifting the economy from high-speed development to high-quality development. While actively introducing highly skilled personnel, enterprise managers can start from many aspects, such as the reform of the forestry enterprise system, the optimization of the working environment of employees and improving the treatment of employees to effectively ensure the retention of talents within the forestry enterprise, and investing in certain material and financial resources in the training of talents to help achieve sustainable development for forestry enterprises. In addition, the factors affecting value chain enhancement include international trade, technological innovation, industrial policy, etc. Therefore, managers should pay attention to enterprise technology innovation and focus on improving the level of technology and equipment, while focusing on market innovation and grasping the direction of technology innovation choices.

(4) According to the model results, "supply chain synergy" is an effective way to achieve the efficient development of enterprises in the future. In addition to the economic

cost savings compared with traditional supply chains, decision making in the scenario of "supply chain synergy" is highly adaptable and can be applied to different dimensions of strategic objectives. Managers can build a "supply chain synergy" development model from the above aspects. In addition, under the continuous improvement of government subsidy policies and the industry environment, corporate decision makers should also take the standpoint of establishing an acceptable balance between sustainability dimensions and appropriately ignore small economic benefits in order to reduce the harmful effects of production processes on the environment.

## 6. Conclusions

Based on previous studies, this study takes the paper and paper products manufacturing industry as an example, and considers the multi-objective optimization of the forest industry under the action of three chains (supply chain, industry chain, and value chain), which makes the study more realistic. Industry chain extension, value chain improvement, and "supply chain synergy" can reduce the economic cost of enterprises, especially the "supply chain synergy" scenario, which has a more obvious effect on economic cost reduction. From the viewpoint of pursuing profits, enterprise managers are more eager to realize "supply chain collaboration". However, the development of "supply chain synergies" seems undesirable from a sustainability performance perspective, as low economic costs are accompanied by increased negative environmental impacts. It should be noted that this model examines the performance of processing firms under a sustainable closed-loop supply chain from the firm's perspective, and does not yet take into account the benefits at all levels of the upstream firm's supply chain network. If "supply chain synergy" can be effectively realized, the carbon emissions of upstream enterprises such as suppliers in raw material production and transportation will be reduced, the overall environmental benefits of the whole supply chain network will be changed, and the final sustainable performance level will be further measured.

Overall, the supply chain design and planning model presented in this paper can be applied to the forestry industry. It should be noted that the model has some limitations when used for other national and regional applications. For example, social objective modeling can only be used when hiring rather than layoffs are necessary, and environmental and cost objective modeling must be revised when multi-product, multi-stage, and multi-production mode decision problems are taken into account. Robust optimization is used in the model to deal with uncertainty, and stochastic programming techniques can be used to deal with the uncertainty parameters when more historical data on market fluctuations are available.

In addition, in future studies, the algorithm can be improved to obtain more accurate values or to consider more factors for sensitivity analysis from other aspects, such as the selection of model coefficient factors. At the same time, as the research on "supply chain synergy" continues to be enriched, the scenario parameters proposed in the paper can be further improved to provide more comprehensive and in-depth theoretical support for the high-quality development of the industry.

**Author Contributions:** Conceptualization, S.W. and X.T.; Investigation, S.W.; Methodology, X.T.; Writing—original draft, S.W.; Writing—review & editing, S.W. and X.T. All authors have read and agreed to the published version of the manuscript.

**Funding:** This research was funded by National Social Science Fund of China (BIA190199).

**Data Availability Statement:** Not applicable.

**Acknowledgments:** We thank openbiox community and Hiplot team (https://hiplot.com.cn) for providing technical assistance and valuable tools for data analysis and visualization.

**Conflicts of Interest:** The authors declare no conflict of interest.

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
