# Peer review of "Research on Sustainable Closed-Loop Supply Chain Synergy in Forest Industry Based on High-Quality Development: A Case Study in Northeast China"

_forests, doi:10.3390/f13101587_

Round 1

Reviewer 1 Report

The manuscript generally is in good appearance. It sounds scientifically and is interesting for the reader. There are some points that can be improved in order the final paper to be topical not only for the scientists that want to cite something on this thematic, but also for the practitioners who would like to use the results. My remarks are following:

1. The empirical data should be described a little better. In the rows between 403-443 there are parameters of the enterprises, which are too numerical. It is not clear the time period they have been averaged.

2. Are the chosen enterprises enough to be representative for the methodology? The results are too particular to the research objects but not in general and probably is appropriate to outlined in the title than the approach have been implemented in particular circumstances to a particular time period.

Wish you good luck with the paper

Reviewer 2 Report

Dear Authors

the paper sounds good and it might have a wide interest among the researchers dealing with supply chain management. I suggest few changes as follow:

Row 33: please add the geographical reference that I think is China or Chinese forests

Row 34-40: I suggest to state explicitly the meaning of sustainability for the paper, because there are several meanings which include economic, ecologic and social sustainability. You report a sustainability concept that focus on the sole timber production sustainability, while in row 56-57 you state a more wider sustainability concept. Please rephrase the text and give more explicit emphasis on what you mean about sustainability at the beginning.

Row 125-126: incorrect reference. Please control and correct.

Row 132-132: incorrect reference. Please control and correct.

Row 588-589: how is it possible? It’s practically unreliable. Could you comment on it or give more evidence about the constant trend of f_soc function?

Another general comment is about the absence of a discussion on the application of the model in other countries? or the limits of the model to be used in countries where there are several constrains on the use of the listed variables. My concern is the difficulties you might have on the application of your model in different economic, ecological and social "environments".
Looking forward on your improvements.

Best regards
